# Cabazitaxel-Loaded Nanocarriers for Cancer Therapy with Reduced Side Effects

**DOI:** 10.3390/pharmaceutics11030141

**Published:** 2019-03-25

**Authors:** Nagavendra Kommineni, Shaheen Mahira, Abraham J. Domb, Wahid Khan

**Affiliations:** 1Department of Pharmaceutics, National Institute of Pharmaceutical Education & Research (NIPER), Hyderabad 500037, India; nagavendra.kommineni@gmail.com (N.K.); shaheen.777.phm@gmail.com (S.M.); 2School of Pharmacy-Faculty of Medicine, The Hebrew University of Jerusalem, Jerusalem 91120, Israel; avid@ekmd.huji.ac.il

**Keywords:** Jevtana^®^, breast cancer, cabazitaxel, liposomes, nanoparticles, pharmacokinetics

## Abstract

Jevtana^®^ is a micellar cabazitaxel (CBZ) solution that was approved for prostate cancer in 2010, and recently, this drug has been reported for breast cancer. The purpose of this study is to evaluate the mediated delivery of CBZ via liposomes and nanoparticles (NPs) for the treatment of breast cancer and compare these with a micellar formulation that is currently in clinical use. CBZ-loaded nanocarriers were prepared with particle sizes between 70–110 nm, and with the sustained in vitro release of CBZ for more than 28 days. Cytotoxicity studies on MCF-7 and MDA-MB-231 cells demonstrated the toxic potential of these nanocarriers. Cellular internalization revealed that NPs and liposomes have better permeability than micelles. Cell cycle analysis and apoptosis studies on MCF-7 and MDA-MB-231 cells confirmed G2/M phase arrest as well as cell death due to apoptosis and necrosis, where formulations were found to be effective compared to a micellar CBZ solution. Results from pharmacokinetic studies revealed that there is an increased circulation half-life and mean residence time for CBZ liposomes and NPs in comparison with a micellar CBZ solution. CBZ liposomes and NPs showed a reduction in hemolysis and neutropenia in comparison with a micellar CBZ solution in rats.

## 1. Introduction

Breast cancer is the most prevalent cancer in women, causing an increased rate of mortality in most countries. However, there are remedies for disease, but chances of relapse in disease progression is frequently observed [1]. Multidrug resistance is the major problem in the failure of many forms of chemotherapy [2]. Other major problems with chemotherapeutics that are available in the market are producing serious side effects leading to death. Thus, there is an extreme need of innovative medication such as nanotechnology-based chemotherapeutics to treat multidrug-resistant breast cancer [3,4]. There is a need for more appropriate strategies, and their implementation will prove highly challenging in optimal clinical management of breast tumors [5].

Taxanes are a class of diterpenes (Paclitaxel (PTX), Docetaxel (DTX), Cabazitaxel (CBZ), Larotaxel) that are microtubule inhibitors widely used as chemotherapeutic agents for several types of cancers [6]. CBZ was USFDA approved in the year 2010 to treat hormone refractory prostate cancer by the name Jevtana^®^. It is a surfactant (polysorbate 80)-based micellar formulation composed of ethanol as a co-solvent to improve the solubility of CBZ and given as an intravenous infusion [7]. The major drawbacks of the marketed formulation are neutropenia (94%), anemia (97%), and thrombocytopenia (47%). Neutropenic fever was observed in 8% of patients. Other common toxicities are diarrhea (47%), nausea (34%), vomiting (22%), hypersensitivity syndrome, bronchospasm, etc. The USFDA has issued a black box warning letter due to severe neutropenia in the patients. All of these side effects, which are life threatening, had inevitably limited the usage of CBZ in clinical studies. Moreover, the CBZ marketed formulation lacks the tumor-targeting ability, and its uneven distribution in the body causes toxicity to normal tissues. Extensive plasma protein binding and the rapid elimination of CBZ needs a frequent dosing schedule, which is lethal to the patient [7]. CBZ has potential activity in metastatic breast cancer, which is under phase II clinical trials [8]. With this, a multidose-escalating study of CBZ in combination with capecitabine in patients with metastatic breast cancer progressing after anthracycline and taxane treatment, which is under phase I/II study [9,10]. Similarly, CBZ has a potential antiproliferative activity in breast cancer in comparison to parent compound DTX, due to the strong suppression of microtubule dynamics, faster drug uptake, and better intracellular retention [11]. 

In recent studies, the therapeutic efficacy and anti-tumor potential of CBZ-loaded polymeric micelles were studied on 4T1 breast cancer cells in an orthotopic metastatic breast cancer model. A 71.6% inhibition in tumor growth and 93.5% reduction of lung metastasis was observed [12]. With this, the enhanced anti-tumor activity of hyaluronic acid surface functionalized solid lipid NPs of CBZ were studies for CD44+ receptor targeting in breast cancer (MCF-7) cell lines, there was an enhanced cellular penetration, cytotoxicity was observed in comparison to drug solution [13]. Tumor-targeted albumin NPs for the co-delivery of CBZ and indocyanine green with irradiation were studied; with this, there was an improved antiproliferative ability (83.5%), and this achieved a higher tumor inhibition rate (91.3%) and improved anti-tumor efficacy in a 4T1 tumor-bearing mice model [14]. Bryan et al. studied the effect of CBZ-conjugated nanoparticles for DTX-resistant and bone metastatic prostate cancer. Cellax–CBZ delivery to the tumor was enhanced and cured 70% of mice bearing bone metastases of resistant prostate tumor, and they also reported mild neutropenia in animals [15]. However, there is no report available for the comparative studies presented in this manuscript. 

Poly(lactic-*co*-glycolic acid) (PLGA) is a United States Food and Drug Administration (FDA)-approved biodegradable polymer, which has been extensively explored as a drug carrier for cancer-targeted therapy [16,17]. Lipid-based fatty bubbles or liposomes are the membrane mimetic biodegradable and biocompatible drug carriers for several diseases [18]. In this study, we prepared CBZ-loaded PLGA nanoparticles and PEGylated-liposomes for breast cancer therapy, as well as to reduce the toxicity of the marketed surfactant-based formulation.

In this work, we have developed surfactant-free nanoformulations of CBZ to address toxicity issues related to micellar CBZ solution for breast cancer. This study compares the biodegradable lipid and polymer-based nanocarriers with surfactant-based micellar CBZ solution for effective delivery in breast cancer. Identifying the therapeutic potential of CBZ-loaded nanocarriers in a comparative manner for the treatment of breast cancer is entirely new for drug delivery.

## 2. Materials and Methods

### 2.1. Materials

Hydrogenated soya phosphatidylcholine (HSPC) and 1,2-distearoyl-*sn*-glycero-3-phosphoethanolamine-*N*-[carboxy(polyethylene glycol)-_2000_] (sodium salt) (DSPE-PEG-_2000_) (Lipoid, Ludwigshafen am Rhein, Germany) were obtained as a gift sample from Cipla Ltd., (Mumbai, India), CBZ was a gift sample from TherDose Pharma Pvt. Ltd. (Telangan, India), *N*-[1-(2,3-Dioleoyloxy)propyl]-*N*,*N*,*N*-trimethylammoniumchloride (DOTAP), 3β-[*N*-(Dimethylaminoethane)-carbamoyl]-cholesterol (DC-Chol), 1,1′,2,2′-tetramyristoyl-cardiolipin, 1-Oleoyl-2-hydroxy-*sn*-glycero-3-phosphocholine (LPC), and 1-Oleoyl-2 hydroxy-*sn*-glycero-3-phosphate sodium salt (OHGP) (Avanti polar lipids, Alabaster, AL, USA) were obtained as gift samples from Dr. Reddy’s Laboratories Ltd. (Hyderabad, India). Cholesterol (Chol) was obtained from Lobachem (Mumbai, India), PLGA (50:50) with a molecular weight of 30,000 to 60,000 Da, poly (vinyl alcohol) (PVA), and Pluronic F-68, 3-(4,5-dimethylthiazol-2-yl)-2,5-diphenyltetrazolium bromide (MTT), rhodamine, fluorescein isothiocyanate (FITC), trypsin EDTA, and cellulose dialysis tubing with 12,000-Da molecular weight cut off were purchased from Sigma Aldrich (St. Louis, MO, USA). Didodecyltrimethylammonium bromide (DMAB) was purchased from Alfa Aesar (Ward Hill, MA, USA). Organic solvents including acetonitrile, dichloromethane (DCM), ethyl acetate, chloroform, and the methanol of HPLC grade, tween 80 were purchased from Merck Pvt. Ltd. (Mumbai, India). Cell culture plates were purchased from Corning Inc. (Corning, NY, USA). The cell culture media and supplements including Dulbecco’s modified Eagle’s medium high glucose (DMEM), fetal bovine serum (FBS), l-glutamine, and antibiotic solutions were purchased from Life Technologies, Inc. (Carlsbad, CA, USA). Amicon ultra 10 kDa cut off centrifugal filters were purchased from Merck Millipore (Darmstadt, Germany).

### 2.2. RP-HPLC Method for Quantification of Cabazitaxel

The analytical method of CBZ was developed by the RP-HPLC (Waters, Milford, MA, USA) method [19,20,21]. CBZ was dissolved in methanol to obtain a stock solution of 1 mg/mL. A series of standard solutions were prepared in concentration range of 1 μg/mL, 5 μg/mL, 10 μg/mL, 20 μg/mL, 40 μg/mL, 60 μg/mL, 80 μg/mL, and 100 μg/mL. Sample (10-µL) injections were injected, and CBZ was separated using Inert Sustain^®^ C18 column (150 × 4.6 mm, 3.5 μm). Mobile phase of acetonitrile: 0.1% formic acid in millipore water (70:30 *v*/*v*) was used in isocratic mode at a flow rate of 1 mL/min. The sample was monitored using a PDA detector (waters 2998, Milford, MA, USA) in which the detection wavelength was set at 230 nm.

### 2.3. Preparation and Optimization of CBZ Liposomes

Blank and CBZ liposomes were prepared by the thin film hydration method using a rotary evaporator. HSPC, Chol, and DSPE-PEG-_2000_ at a 5.5:2.5:1 weight ratio were dissolved, and the thin film was formed according to the previously reported method [4] with slight modification. Briefly, the lipids and drug were dissolved in 6 mL of chloroform/dichloromethane in a round-bottomed flask. Then, the thin film was prepared using a rotary evaporator (Hei-VAP unit, Heidolph, Germany) at 37 °C, at 140 rpm under vaccum to evaporate the solvent for 30 min. Preformed uniform films were hydrated above the phase transition temperature of the lipids with the help of hydration media followed by sonication (20-s pulse on and 10-s pulse off cycles at 40% amplitude) (Sonics and Materials Inc., Newtown, CT, USA) to get uniform liposomal dispersion. The effects of different parameters such as the solvent, sonication time, lipid concentration, type of lipids, hydration media, and percentage of drug loading were varied to obtain optimum CBZ-loaded liposomes. 

### 2.4. Preparation and Optimization of CBZ Nanoparticles

CBZ-loaded PLGA NPs were prepared by the emulsion solvent diffusion technique, which was previously employed in our lab [22,23] with slight modification. PLGA and CBZ were dissolved in 2 mL of ethyl acetate to get 4% (*w*/*v*) organic phase. This polymer drug solution was added drop wise to 4 mL of water phase-containing 0.5% (*w*/*v*) surfactant, and then homogenized (Ultra Turrax T25, IKA Germany, Königswinter, Germany) at 15,000 rpm for 5 min. The obtained primary emulsion was added to 24 mL of 0.05% surfactant solution and stirred for 3 to 4 h to allow solvent evaporation. CBZ-loaded polymeric NPs were recovered by centrifugation (Sigma Laborzentrifugen GMBH, Osterode am Harz, Germany) at 20,000 rpm for 30 min. Nanoparticle pellet was redispersed in water; the centrifugation and redispersion process was repeated three times to ensure the removal of surfactant before further characterization. The specified parameters in the preparation of NPs such as type of surfactant, surfactant concentration, and percentage of drug loading were optimized by keeping other parameters constant according to an earlier reported method [22].

### 2.5. Characterization of Liposomes and Nanoparticles

#### 2.5.1. Particle Size Distribution

The mean particle size (z-average), polydispersity index [24], and zeta potential of CBZ-loaded liposomes and NPs were measured by dynamic light scattering technique at 25 °C with 90° scattering angle using a Malvern Zeta sizer Nano (Malvern Instrument Ltd., Malvern, UK). All of the prepared formulations were diluted with millipore water at 1:10 dilution and analyzed in triplicate.

#### 2.5.2. Drug Loading and Entrapment Efficiency

Theoretical drug loading was varied from 5% to 20% *w*/*w* for liposomes and NPs. CBZ entrapment efficiency was determined by an ultrafiltration method. Briefly, 1 mL of liposomes/NPs dispersion was placed into an Amicon Ultra 4 centrifugal filter unit with a nominal molecular weight cut off of 10 kDa (Merck Milipore Ltd., Darmstadt, Germany) and centrifuged at 10,000 rpm for 10 min [25]. The free drug present in filtrate was measured by RP-HPLC (described in Section 2.2). The amount of drug entrapped was obtained by subtracting the amount of free drug from the total drug incorporated in 1 mL of liposomes or NPs’ dispersion. The percentage entrapment efficiency (% EE) was calculated using Equation (1).
(1)$$\%\;\mathrm{EE}=\frac{\mathrm{Amount}\;\mathrm{of}\;\mathrm{drug}\;\mathrm{entrapped}}{\mathrm{Total}\;\mathrm{amount}\;\mathrm{of}\;\mathrm{drug}\;\mathrm{taken}}\times 100$$

#### 2.5.3. Transmission Electron Microscopy

The size and surface morphology of liposomes and NPs were determined using transmission electron microscopy [26]. CBZ-loaded liposomes and NPs were deposited on copper grids pre-coated with carbon and dried for 5 to 10 min. The grids with preloaded formulations were stained with 2% *w*/*v* of uranyl acetate (negative staining), washed, air dried, and images were captured using JEM 2100 transmission electron microscope (JOEL, Tokyo, Japan) with an accelerating voltage of 120.0 kV using Digital Micrograph^®^ software (Gatan, Inc., San Francisco, CA, USA). Analysis was performed at 25 ± 2 °C.

### 2.6. In Vitro Drug Release

In vitro drug release was determined by the previously employed dialysis bag method [27,28,29]. Jevtana^®^ consists of solubilized CBZ in polysorbate 80, and 13% *v*/*v* ethanol was used as a reference to compare the release profile of CBZ liposomes and NPs. Equal concentrations of 1 mL of micellar CBZ solution/liposomes/NPs dispersion and 1 mL of PBS (pH 7.4) with 0.5% *w*/*v* tween 80 were placed in a preactivated dialysis membrane with a 12-kDa molecular weight cut off. Then, the closed bag was immersed in 30 mL of release media (PBS of pH 7.4 containing 0.5% *w*/*v* tween 80). The tubing was kept in a shaker bath, with 120 rpm at 37 °C. Release samples (2 mL) were periodically removed at predetermined time intervals for analysis, and replaced with the same volume of fresh medium in order to maintain the volume. Collected samples were analyzed for drug content by HPLC, as described in a previous Section 2.2. Data obtained in triplicate was analyzed graphically and expressed as percentage of cumulative drug release versus time. Further in vitro drug release data obtained was fitted into zero-order, first-order, Higuchi, and Peppas models to study the drug release kinetics from the developed formulation.

### 2.7. Cell Viability on MCF-7 and MDA-MB 231 Cells

Breast cancer cell lines (MDA-MB-231 and MCF-7) are used in the current study to know the cytotoxicity potency of formulations. MDA-MB-231 cell lines are resistant and triple-negative breast cancer cell lines, whereas MCF-7 cell lines are sensitive and hormone receptor-positive. The cell lines are procured from the National Centre for Cell Science, Pune and were cultured in Dulbecco’s modified Eagle’s medium (DMEM) containing 10% heat-inactivated fetal bovine serum (FBS), 1.5 g/L of NaHCO_3_, 2 mM of l-glutamine, 10,000 units of penicillin, 10 μg/mL of streptomycin, and 25 μg/mL of amphotericin B, incubated at 37 °C with 5% CO_2_ in a humidified atmosphere. The cells were seeded (6500 cells/well) and incubated overnight. Cells were treated with CBZ micellar solution, and developed formulations (CBZ NPs, CBZ liposomes) at concentrations of 1 µM, 10 µM, and 100 µM for 48 h. After treatment, MTT solution (500 μg/mL) was added and incubated for 3 h. Then, DMSO was added (200 μL) into each well to dissolve the formazan crystals. The optical density was measured using a microplate reader (Spectramax M4, Molecular Devices, San Jose, CA, USA) at 570 nm. Cell viability (%) was calculated as the ratio of the number of living cells in treated samples to that of the control [15,30]. All of the experiments were performed in triplicate, and the results were represented as mean ± SD.

### 2.8. Cellular Internalization of Nanocarriers by Confocal Microscopy

Briefly, MDA-MB-231 cells were seeded in six-well plates containing sterile cover slips and incubated for 24 h. Cells were treated with FITC-loaded formulations (FITC-Micelles, FITC-NPs, FITC-Liposomes) at 100-µM concentration dispersed in culture medium for 3 h, and washed several times with PBS. Then, cells were treated with rhodamine solution for 30 s as a secondary dye and washed with PBS. Cover slips were mounted with VECTASHIELD HardSet antifade mounting medium^®^ with DAPI (Vector Labs, Burlingame, CA, USA) on a glass slide. Slides were visualized using a confocal microscope (Leica TCS SP8 Laser Scanning Spectral Confocal) [16,31].

### 2.9. Inhibition of Cell Cycle and DNA Damage Checkpoints

MCF-7 and MDA-MB-231 cells were seeded in six-well plates at a density of 1 × 10^5^ cells/mL and incubated for 24 h. Cells were treated with different formulations for 48 h to know the effect on cell cycle. After the treatment period, cells were washed with PBS, harvested using trypsin EDTA, fixed in 70% *v*/*v* chilled ethanol, and stored overnight at 2 to 8 °C. Fixed cells were pelleted, resuspended, and treated with RNAse A for 15 min to get rid of RNA. Then, cell nuclei were stained by propidium iodide reagent (400 μg/mL) and incubated for 30 min at 37 °C in the dark [32,33]. Flow cytometric analysis (BD FACSVerse™, Franklin Lakes, NJ, USA) was performed by doublet discrimination module with 10,000 events. 

### 2.10. Cytometric Detection of Apoptosis

MCF-7 and MDA-MB-231 cells (1 × 10^6^ cells/well) were seeded in six-well culture plates and allowed to attach for 24 h. Cells were treated with CBZ solution, CBZ liposomes, and CBZ NPs at 100-µM concentration for 48 h. Afterwards, incubation cells were trypsinized and centrifuged at 1000 rpm for 5 min. The extent of apoptosis was determined using the Annexin V-FITC and PI dead cell apoptosis kit (Molecular Probes, Thermo Fisher Scientific, Waltham, MA, USA) as per manufacturer’s protocol, and the treated cells were analyzed using a flow cytometer (BD FACSVerse™, Franklin Lakes, NJ, USA) [17,34].

### 2.11. Ex Vivo Hemolysis

The hemolytic potential of the micellar CBZ solution, CBZ NPs, and CBZ liposomes were determined according to a previously employed method [35,36] with slight modification. Blood samples were collected from the rat in heparinized tubes and centrifuged at 5000 rpm for 5 min. The supernatant was discarded, and the erythrocyte suspension was washed with normal saline two to three times to remove lysed hemoglobin. Erythrocyte concentration was prepared by diluting 1 mL of blood to 50 with normal saline. A stock of CBZ solution and liposomes and NPs were prepared at concentrations of 200 µg/mL, 100 µg/mL, 50 µg/mL, 10 µg/mL, 1 µg/mL, 0.1 µg/mL. Triton X100 (20% *w*/*v*) acted as positive control (100% lysis), and normal saline acted as negative control (0% lysis) with mean ± SD, *n* = 3. For every 900 µL of erythrocyte suspension, 100 µL of test sample was added, and then incubated for 1 h at 37 °C. Then, the above samples were centrifuged at 5000 rpm for 10 min at 4 °C. Collected supernatant was analyzed with the help of a microplate reader at 541 nm. Percentage hemolysis was calculated using following equation (Equation (2)):
(2)$$\%\;\mathrm{Hemolysis}=\frac{\mathrm{Abs}\;\mathrm{of}\;\mathrm{test}\;\mathrm{sample}-\mathrm{Abs}\;\mathrm{of}\;\mathrm{negative}\;\mathrm{control}}{\mathrm{Abs}\;\mathrm{of}\;\mathrm{positive}\;\mathrm{control}-\mathrm{Abs}\;\mathrm{of}\;\mathrm{negative}\;\mathrm{control}}\times 100$$

### 2.12. In Vivo Studies

In vivo animal experiments were carried out on female Sprague–Dawley (SD) rats (Teena Laboratories, Hyderabad, India) of 180 to 200 g weight in accordance with the guidelines of CPCSEA, India. The animal protocol (NIP/8/2017/PE/240) was approved by the Institutional Animal Ethics Committee (IAEC) of the National Institute of Pharmaceutical Education and Research, Hyderabad, India. Animals were housed under controlled conditions of light (12 h of light and 12 h of dark) at 22 ± 2 °C and 50 ± 15% relative humidity. Animals were supplied with a standard diet, water ad libitum, and acclimatized to laboratory conditions for one week prior to experiments.

#### 2.12.1. Neutropenia in Animals

Jevtana^®^ is a marketed surfactant-based micellar CBZ solution that produced severe neutropenia and anaphylaxis with bone marrow toxicity [37]. In this study, prepared liposome and NPs of CBZ were injected to female SD rats (*n* = 5) by i.v. route at a dose of 2.5 mg/kg and compared with micellar CBZ solution at the same dose along with saline control. Blood samples were collected at intervals of 1, 2, 4, 7, 9, 12, and 14 days, and analyzed for blood cell parameters using an ADVIA 2120 hematology analyser (Siemens, Washington, DC, USA) [38].

#### 2.12.2. Pharmacokinetics

Pharmacokinetic study was conducted to evaluate the fate of drug administered in the form of liposomes, NPs, and micellar CBZ solution via an intravenous route of administration. Female SD rats were selected for the study (*n* = 6), and a 15 mg/kg dose was administered. Blood samples were collected at different intervals of time (5 min, 15 min, 30 min, 45 min, 1 h, 2 h, 3 h, 6 h, 24 h, and 48 h) through a retro orbital route. Blood samples were centrifuged at 6000 rpm at 4 °C. The obtained plasma samples were extracted by the protein precipitation method and analyzed with the help of the RP-HPLC method. A calibration curve was constructed in the range of 100 to 20,000 ng/mL, and found to be linear with a correlation coefficient of 0.9998. The sample preparation procedure followed for the calibration curve was 10 µL of working stock solution of CBZ, and 50 µL of IS Paclitaxel (PTX) was added to 140 µL of plasma matrix and vortexed on a shaker for 2 min. To the above mixture, 300 µL of acetonitrile and methanol (50:50) were added as precipitating solvents, vortexed for 5 min, and centrifuged for 10 min at 6000 rpm. Supernatant was collected and injected to RP-HPLC [39]. Plasma and tissue pharmacokinetic parameters were calculated from Kinetica–Thermo Fischer Scientific Software. The linear trapezoidal rule was used to calculate the area under the plasma mean concentration time curve. *C*_max_ values were obtained from a visual inspection of the obtained data.

### 2.13. Statistical Analysis

Analysis was carried out using GraphPad Prism 6.0 (version 6.05, GraphPad Software Inc., La Jolla, CA, USA). Statistical significance was determined using two-way ANOVA followed by Bonferroni’s multiple comparisons test; pharmacokinetic parameter statistical significance was determined using one-way ANOVA followed by Tukey’s multiple comparison tests, where *p* < 0.05 was considered statistically significant, and *p* < 0.01 was considered highly significant. Data is expressed as the mean ± SD.

## 3. Results and Discussion

### 3.1. Preparation and Optimization of CBZ Liposomes

Particle size is an important parameter for the performance of any nanotherapeutics, as it alters the pharmacokinetic and pharmacodynamics behavior of the drug. Nanotherapeutics with a size less than 200 nm can be easily accumulated in tumor tissue by EPR effect, whereas the larger particles with a size greater than 200 nm are prone to non-specific systemic elimination by reticuloendothelial cells and phagocytes [31,40,41]. CBZ-loaded liposomes were prepared by the thin film hydration method using a rotary evaporator followed by sonication to get uniform liposomal dispersion. Optimization studies were performed initially for the preparation of blank liposomes. 

For the optimization of sonication time, blank liposomes were first prepared with different sonication times (5 min, 10 min, 15 min, and 20 min) using two solvents (chloroform and dichloromethane (DCM)). Other parameters such as the type of lipid HSPC and lipid concentration (1.8 mg/mL) were kept constant. Particle size and the PDI of obtained liposomes were determined. Liposomal dispersion prepared with chloroform (Figure 1A) as the solvent had shown the lowest particle size (94.6 ± 3.1 nm) and PDI (0.271 ± 0.04) with 10 min of sonication, so chloroform was selected for the further development of CBZ-loaded particles. 

For the optimization of lipid concentration, blank and CBZ liposomes were prepared with different HSPC concentrations (0.9 to 9 mg/mL) by keeping other parameters constant (sonication time of 10 min and 10% *w*/*w* drug loading). Once obtained, all of the batches of blank liposomes were below 120 nm in size with a PDI less than 0.4 (Figure 1B). However, in the drug-loaded liposomes, an increase in lipid concentration and a gradual increase in the particle size and PDI were observed. The CBZ-loaded formulation with 1.8 mg/mL showed a particle size (92.07 ± 2.01 nm) with a PDI of 0.241 ± 0.03, and this lipid concentration was selected for further optimization. 

Further, different types of lipid (HSPC, DOTAP, DC Chol, cardiolipin, LPC, and OHGP) at 1.8-mg/mL concentration were studied. All of the batches were prepared and the particle size, PDI, and entrapment efficiency were checked by keeping other parameters constant (10% *w*/*w* CBZ loading, 10-min sonication). DOTAP has given the least particle size (93.62 ± 1.91 nm) with a PDI of 0.32 ± 0.036 with 79.63 ± 4.55% entrapment efficiency (Figure 1C) when compared to other liposome formulations. With HSPC, we obtained particle size (111.63 ± 1.27 nm), PDI (0.189 ± 0.008), and 89.55 ± 0.45% entrapment efficiency. Hence, we selected HSPC liposomes for further studies, considering their higher entrapment efficiency. Zeta potential plays a critical role in the stability of formulation if they are intended for extended storage conditions. Particles should have neutral or slightly negative zeta potential in order to minimize the nonspecific binding to endothelium [42]. The comparative zeta potential of prepared liposomes was also determined and represented in Figure 1D. DOTAP (42.37 ± 0.45 mV) and DC Chol (35.6 ± 0.87 mV) liposomes have positive zeta potential due to the cationic nature of the lipids, but others are anionic HSPC (−21.43 ± 0.32 mV), cardiolipin (−12.33 ± 1.62 mV), LPC (−7.19 ± 1.69 mV), and OHGP (−49.53 ± 3.02 mV). HPSC is the component of FDA-approved Doxil^®^ liposomes with a slightly negative zeta potential and prolonged storage stability of about two years [43]. So, based on particles’ size and surface charge, HSPC liposomes were selected for further studies.

For the optimization of CBZ loading in HSPC liposomes, different batches of liposomes with varied CBZ loading (5–20% *w*/*w*) were prepared, and the particle size distribution, zeta potential, and entrapment efficiency were determined for all of these batches. Particle size for 10% *w*/*w* drug-loaded liposomes was 92.09 ± 1.056 nm with a PDI of 0.237 ± 0.01 and 90.89 ± 1.02% entrapment efficiency (Figure 1E). Whereas, the 15% and 20% *w*/*w* drug-loaded liposomes showed a drastic increase in particle size (223.3 ± 3.21 and 236.1 ± 3.38 nm, respectively) and PDI values of 0.634 ± 3.21 and 0.501 ± 3.38, respectively. At different drug loadings, there was not much difference in the zeta potential (−24.4 ± 2.34 mV, −29.9 ± 1.58 mV, −29.3 ± 2.12 mV, and −30.6 ± 3.39 mV) of HSPC liposomes (Figure 1F). So, the optimized liposomes with 10% *w*/*w* CBZ loading were selected for further studies.

For the above-optimized liposomes, the effect of different hydration media such as 5% *w*/*v* mannitol solution, 5% *w*/*v* dextrose solution, 10% *w*/*v* sucrose solution, 0.9% *w*/*v* saline solution, and PBS (pH 7.4) were checked for the particle size distribution, percentage of entrapment efficiency (Figure 1G), and zeta potential (Figure 1H). The formulation with 10% *w*/*v* sucrose solution showed the most desirable particle size (108.53 ± 1.5 nm) with a PDI of 0.257 ± 0.04, entrapment efficiency of 87.63 ± 0.3% and zeta potential of −24.35 ± 1.2 mV, and was thus selected for further studies. Thus, the optimized CBZ liposomes batch was prepared with HSPC as the selected lipid, chloroform as a solvent, 10% *w*/*v* sucrose solution as the hydration medium, a 10-min sonication time, and 10% *w*/*w* CBZ loading, and selected for further studies.

### 3.2. Preparation and Optimization of CBZ Nanoparticles

CBZ NPs were prepared by the emulsion solvent diffusion technique. Optimization studies were carried out with 10% *w*/*w* CBZ loading. Different surfactants such as PVA, pluronic F-68, and DMAB at 0.5% *w*/*v* concentration were first optimized. CBZ NPs with DMAB as a surfactant showed the smallest particle size (91.6 ± 1.01 nm) with a PDI of 0.152 ± 0.008 and entrapment efficiency of 96.95 ± 0.56% (Figure 2A). Further, DMAB concentration was varied (0.25% to 1% *w*/*v*) with different drug loading (10% to 20% *w*/*w*). From the results, it was observed that with 0.25% and 0.5% *w*/*v* concentrations of DMAB, there was a slight difference in the particle size and PDI (Figure 2B,C), and a 0.5% *w*/*v* concentration of DMAB with 20% *w*/*w* CBZ has shown a particle size of 76.19 ± 3.54 nm and PDI of 0.19 ± 0.021.

However, at 1% *w*/*v* of DMAB concentration, a smaller particle size (77.31 ± 7.7 nm) with a slight increase in the PDI (0.345 ± 0.02) was observed (Figure 2D). With this, the percentage of entrapment efficiency at different loading was almost similar (98.35 ± 1.02%, 98.68 ± 0.71%, and 98.81 ± 1.16%). However, the higher concentration of surfactant (1% *w*/*v*) may cause toxicity, so 0.5% *w*/*v* DMAB concentration with 20% *w*/*w* CBZ loading was optimized for further studies. The zeta potential for CBZ NPs at different surfactant concentrations with different loading is shown in Figure 2E. The final optimized CBZ NPs with 0.5% *w*/*v* DMAB as a surfactant and 20% *w*/*w* CBZ loading were selected for further studies.

### 3.3. Size, Size Distribution, Zeta Potential, and Entrapment Efficiency of Optimized CBZ Liposomes and Nanoparticles

In an optimized CBZ liposomes batch, an average particle size of 108.53 ± 1.5 nm (Figure 1I), with a PDI of 0.257 ± 0.04 was observed. Surface charge distribution was found to be −24.35 ± 1.2 mV due to the negative charge of HSPC. The entrapment efficiency of optimized CBZ-loaded HSPC liposomes was found to be 87.63 ± 0.3%. Optimized CBZ NPs have a size of 76.19 ± 3.55 nm (Figure 2F), with a PDI of 0.19 ± 0.021 and zeta potential of +45.73 ± 1.65 mV. The entrapment efficiency of optimized CBZ-loaded polymeric NPs was found to be 97.48 ± 1.55%. TEM analysis indicated that the particle sizes of CBZ liposomes and NPs are in the nanometer range with spherical morphology (Figure 3B,C).

### 3.4. In Vitro Drug Release

The in vitro drug release profile of CBZ from liposome dispersion and NPs was compared with micellar CBZ solution. Both liposomes and NPs exhibited a sustained release of CBZ from the dispersion. The CBZ release from the micellar CBZ solution was comparatively fast because of the presence of polysorbate 80 and ethanol. The in vitro drug release data obtained from the micellar CBZ solution, CBZ liposomes, and CBZ NPs were compared for the cumulative drug release. In five days, the micellar CBZ solution showed 98.2 ± 6.08% drug release, whereas the CBZ NPs and CBZ liposomes showed 17.96 ± 5.8% and 45.60 ± 4.29%, respectively. CBZ liposomes and NPs showed 60.98 ± 4.74% and 40.34 ± 8.25% drug releases respectively for 28 days (Figure 3A). The rate of drug release from liposomes and NPs is by diffusion followed by dissolution, as well as an erosion mechanism. From the above results, it can be depicted that the drug diffusion and erosion from the lipid matrix is high when compared to the PLGA matrix. The hydrophobic nature of CBZ and slow degradation of the polymer could be the possible reason for the slow and sustained release. The CBZ micellar solution followed Higuch release kinetics, whereas CBZ NPs and CBZ liposomes followed the Peppas model. The release mechanism of CBZ NPs and CBZ liposomes was determined as non-Fickian diffusion based on the “n” values (0.54 and 0.49) respectively calculated using Peppas equation which was mentioned in the Appendix A (drug release kinetics). 

The release behavior of CBZ liposomes and NPs ensure that the matrix-based systems can maintain the effective concentration for a long time, and have the ability to sustain the treatment to the cancer in vivo.

### 3.5. Cell Viability Assay on MCF-7 and MDA-MB 231 Cells

MTT assay was performed on MCF-7 and MDA-MB 231 cells to determine the extent of the antiproliferative effect of the CBZ micellar solution, CBZ liposomes, and CBZ NPs. Primarily blank formulations were screened, and the percentage of cell viability was found to be 100%. Figure 4A,B presents the percentage of cell viabilities of all of the treated groups after 48 h of incubation. 

Dose-dependent cell inhibition was observed on both the cell lines for all groups and the percentage of cell viability decreased with respect to the concentration of the drug. MCF-7 cells treated with CBZ NPs revealed poor viability (26.20 ± 0.25, 8.57 ± 0.15, and 7.94 ± 0.14) at concentrations of 1 μM, 10 μM, and 100 μM compared with the CBZ liposomes (31.33 ± 0.48, 15.09 ± 0.48, and 10.46 ± 0.12) and CBZ micellar solution (48.81 ± 0.58, 24.42 ± 0.08, and 22.95 ± 0.14) in treated cells. Similarly, MDA-MB-231 cells treated with CBZ NPs revealed poor viability (31.38 ± 0.68, 16.35 ± 0.42, and 12.39 ± 0.75) at concentrations of 1 μM, 10 μM, and 100 μM compared with the CBZ liposomes (28.43 ± 0.27, 26.62 ± 0.72, and 21.11 ± 1.722) and CBZ micellar solution (38.61 ± 1.002, 34.42 ± 0.67, and 30.98 ± 0.21) in treated cells. Among the above treated groups of CBZ liposomes, CBZ NPs were more effective in controlling the growth of cancer cells. From the above results at 100-μM concentration, there was a smaller percentage of cell viability in all of the groups, and hence, we selected the 100-μM concentration for cell cycle analysis and apoptosis study.

### 3.6. Qualitative Cellular Internalization of Nanocarriers by Confocal Microscopy

The majority of anticancer drugs have poor permeability across the cellular membranes. The efficiency of uptake of developed formulations (micelles, NPs, and liposomes) loaded with FITC was qualitatively visualized by CLSM with DAPI as the nuclear stain and rhodamine as the cell membrane stain, as shown in Figure 5. Blue fluorescence was observed as the nuclei were stained with DAPI. The green fluorescence in the cytoplasm was due to formulations loaded with FITC. There was an increase in green fluorescence intensity in cells treated with NPs and liposomes when compared to micelles, which may be due to the rapid uptake of biocompatible and biodegradable nanocarriers by the cells, which will be more beneficial for cancer treatment [33]. Rhodamine was used as a secondary dye to stain cytoplasm. In a merged image, there was an increase in the intensity of yellow color in the cytoplasm of cells treated with FITC-loaded liposomes and NPs, which clarifies that they have more cell internalization capability in comparison to FITC-loaded micelles. However, the yellow color intensity was more dominant in case of cells treated with FITC-loaded liposomes due to rapid internalization through cancer cell membranes.

### 3.7. Inhibition of Cell Cycle and DNA Damage Checkpoints

Flow cytometry analysis was performed to study the distribution of cells in different phases (Sub G1 versus G0/G1 versus S versus G2/M) for the detection of apoptotic cells with fractional DNA content. In this experiment, there was a drastic reduction in the G0/G1 phase arrest in all of the treatment groups, which implies the potent activity of CBZ in different formulations in comparison to control. 

Flow cytograms of groups of MCF-7 cells treated with CBZ solution, CBZ liposomes, and CBZ NPs have respectively shown a 1.6-fold, 2.6-fold, and 2.1-fold increase in G2/M phase arrest when compared to the control group (Figure 6A,A′). Similarly, the flow cytograms of groups of MDA-MB-231 cells treated with CBZ solution, CBZ liposomes, and CBZ NPs have shown 1.6-fold, 3.1-fold, and 2.6-fold increase in G2/M phase arrest when compared to the control group (Figure 6B,B′). The order of G2/M phase arrest observed in MCF-7 and MDA-MB-231 cell lines after treatment are CBZ liposomes > CBZ NPs > CBZ solution. The reason may be due to the enhanced cellular internalization of liposomes and NPs followed by solution in treated cells. Overall, cell cycle analysis data confirmed that CBZ liposomes, CBZ NPs, and CBZ micellar solution were causing apoptosis by causing cell cycle arrest at the G2/M phase. CBZ liposomes outperformed the CBZ micellar solution and CBZ NPs. This will be further confirmed by Annexin V-FITC apoptosis assay.

### 3.8. Cytometric Detection of Apoptosis

The extent of apoptosis induced by CBZ from solution, liposomes, and NPs at 100-μM concentration on both the cell lines was determined by an Annexin V-FITC/PI kit. As a potent inhibitor of microtubules, CBZ acts on the G2/M phase of cell growth cycle, leading to apoptosis, which is then followed by necrosis [44]. The induction of apoptosis or necrosis was investigated in this experiment. The results on MCF-7 cell lines (Figure 7A,A′) demonstrated that the CBZ micellar solution has shown a 3.12-fold induction of apoptosis and 7.61-fold induction of necrosis, whereas the CBZ liposomes have shown an 8.22-fold induction of apoptosis and 5.95-fold induction of necrosis in comparison with control, which implies that the therapeutic potential of liposomes was more when compared to the CBZ micellar solution. CBZ NPs have shown a 30.78-fold induction of necrosis when compared to control; this result is in agreement with the previous reports in which the apoptotic potential of CBZ was analyzed [26]. 

Results on the MDA-MB-231 cell lines (Figure 7B,B′) demonstrated that that CBZ micellar solution has shown a 7.18-fold induction of apoptosis and 3.26-fold induction of necrosis, whereas the CBZ liposomes showed an 11.06-fold induction of apoptosis and a 3.46-fold induction of necrosis in comparison with control, which implies that the anti-tumor potential of liposomes was more when compared to CBZ micellar solution. CBZ NPs have shown a 75.12-fold induction of necrosis when compared to control. From the above results, it can be concluded that the CBZ micellar solution and CBZ liposomes induced apoptosis and necrosis on both the cell lines, whereas the CBZ NPs induced necrosis. From the above results, it was observed that cell death due to apoptosis was more observed in the case of the treatment group of CBZ liposomes rather than the CBZ micellar solution and CBZ NPs, which will be more beneficial for breast cancer therapy.

### 3.9. Ex Vivo Hemolysis

The in vivo activity of CBZ liposomes and NPs will depend on a delicate balance between their selective cytotoxicity toward breast cancer cells while remaining inert toward healthy tissues and cells. After the systemic administration the CBZ micellar solution, CBZ liposomes and NPs will interact with red blood cells (RBCs). Therefore, RBCs hemolysis will dictate the hemocompatibility of the micellar CBZ micellar solution, CBZ liposomes, and CBZ NPs at different concentrations. The obtained results of the CBZ micellar solution, CBZ liposomes, CBZ NPs have shown less than 6% hemolysis at 2500 ng/mL. Although there was a drastic increase in the hemolytic potential of the CBZ micellar solution with the increase in concentration, which was observed up until 20 μg/mL. The results of the percentage of hemolysis at 20 μg/mL for the CBZ micellar solution, CBZ liposomes, and CBZ NPs were observed to be 48.88 ± 1.85, 5.03 ± 0.58, and 13.09 ± 0.56, respectively, as shown in Figure 8A,B.

The above results indicated that CBZ liposomes and CBZ NPs have shown an 8.7-fold and 2.7-fold reduction in hemolytic potential in comparison to the CBZ micellar solution, and they are safe, biodegradable, and biocompatible to administer systemically when compared to a surfactant-based CBZ micellar solution. Furthermore, CBZ liposomes have shown a 1.6-fold reduction in hemolysis in comparison to CBZ NPs at 20 μg/mL.

### 3.10. In Vivo Studies

#### 3.10.1. Neutropenic Study

Jevtana^®^ has a potential toxicity to alter blood cell parameters by reducing the neutrophil count and causing death. So, the toxicity of CBZ liposomes and CBZ NPs in comparison with a surfactant-based CBZ micellar solution was determined. The comparative results are shown in Figure 9. For a single dose of i.v.-administered samples in rats, it was observed that neutrophil count was depleted until one week, after which there was an increase in count. On the fourth day, there was an observed drastic reduction of neutrophil count (Table 1) in this order: CBZ micellar solution (0.1025 × 10^9^/L) < CBZ NPs (1.1 × 10^9^/L) < CBZ liposomes (1.6725 × 10^9^/L) < saline control (2.635 × 10^9^/L). The surfactant-based CBZ micellar solution showed rapid absorption in four days, which lead to neutropenia in animals followed by the elimination of the drug, which sustained the neutrophil count [45]. However, we were not able to get a significant difference in hematological screening between the treatment groups.

The micellar CBZ solution showed a 24.7-fold reduction in neutrophil count in comparison to saline control. However, NPs (1.4-fold) and liposomes (0.6-fold) showed less reduction in neutrophil count in comparison with control. From the above results, it was indicated that NPs and liposomes are the efficient delivery vehicles; they will show sustained release for highly potent and hydrophobic drugs such as CBZ from the biodegradable matrix. Furthermore, CBZ liposomes have shown a lower reduction in neutrophil count in comparison to CBZ NPs. All of the other blood parameters—white blood cell count perox (WBCP), White blood cell count baso (WBCB), RBC, Neut, hemoglobin (HGB), hematocrit (HCT), mean corpuscular volume (MCV), mean corpuscular hemoglobin (MCH), mean corpuscular hemoglobin concentration (MCHC), cell hemoglobin concentration mean (CHCM), red cell distribution width (RDW), hemoglobin distribution width, (HDW), platelets (PLT), and mean platelet volume (MPV)—were the same for all of the formulation treatment groups in comparison to the control (Table 1).

#### 3.10.2. Pharmacokinetics

The mean plasma and tissue pharmacokinetic parameters are shown in Table 2 and Figure 10. *C*_max_ was more for the CBZ micellar solution due to polysorbate 80 and ethanol. Rapid absorption lead to dose dumping and drug-related toxicity such as neutropenia, anaphylaxis, and bone marrow toxicity. The micellar CBZ solution was rapidly eliminated, so there is a need for frequent dosing, which may cause more side effects and drug resistance. Both *C*_max_ and AUC values were less for CBZ NPs and CBZ liposomes in comparison with the CBZ micellar solution. However, the other parameters such as the half-life (0.413-fold and 3.15-fold) and mean residence (0.339-fold and 3.785-fold) time were increased for NPs and liposomes, respectively. It means that they reside in the body for a long period of time, and release the drug in a slow manner with reduced toxicity and dosing frequency. Drastic improvement in the *t*_1/2_ and mean residence time of CBZ liposomes over CBZ NPs was due to the presence of DSPE-PEG-_2000_, which prolongs the circulation and avoids macrophage uptake; this is in correlation with earlier reports [46].

## 4. Conclusions

Efficient nanotherapeutics (liposomes and NPs) were developed for the delivery of highly hydrophobic drug CBZ. The NPs and liposomes that were developed are spherical in shape, with sizes less than 110 nm, which is beneficial for EPR targeting. CBZ liposomes with negative zeta potential (−24.35 ± 1.2 mV) and CBZ NPs with positive zeta potential (+45.73 ± 1.65 mV) may help in stabilizing the formulation. The percentage of drug loading, which was nearly 20% *w*/*w* (entrapment efficiency: ~98%), was higher for CBZ NPs compared with CBZ liposomes, which had 10% *w*/*w* drug loading (entrapment efficiency: ~88%). In vitro drug release studies revealed that there was sustained CBZ release from liposomes (~60%) and NPs (~41%) in 28 days, which is more beneficial for long-term therapy in breast cancer. In vitro cytotoxicity studies on MCF-7 and MDA-MB-231 cell lines revealed that NPs and liposomes are more potent than a CBZ micellar solution; this may be due to the rapid internalization of nanocarriers through the endothelium of the cell lines, which was observed in CLSM images. Cell cycle analysis revealed that CBZ liposomes and NPs have more G2/M phase cell arrest than the CBZ micellar solution on both the cell lines. Apoptosis assay indicated that the CBZ micellar solution and CBZ liposomes caused apoptosis and necrosis, leading to cell death, whereas NPs caused necrosis in both the cell lines. However, CBZ liposomes caused more apoptosis than CBZ NPs and the micellar CBZ solution, which will be more beneficial for breast cancer therapy. The in vitro hemolysis study implied that liposomes (8.7-fold, 5.03%) and NPs (2.7-fold, 13.09%) have less toxicity on RBC in comparison with a surfactant-based micellar CBZ solution (48.88%). Furthermore, CBZ liposomes showed a 1.6-fold reduction in hemolysis in comparison to CBZ NPs at 20 μg/mL concentration. Surfactant-free nanotherapeutics (liposomes and NPs) have shown a 0.6-fold and 1.4-fold reduction in neutrophil count in comparison to control, whereas the micellar CBZ solution showed a 24.7-fold reduction in neutrophil count. CBZ liposomes had a less neutropenic effect than the micellar CBZ solution and NPs. Pharmacokinetic parameters proved that there was an increase in *t*_1/2_ and the MRT of developed formulations, which provided the sustained release of drug, and in turn reduced the dosing frequency. With this, CBZ liposomes have extended *t*_1/2_ (~7.6 h) and MRT (~8.9 h) over CBZ NPs. In conclusion, developed CBZ liposomes can act as potential delivery carriers for the treatment of breast cancer with reduced toxicity. The developed CBZ-loaded NPs and liposomes will be evaluated for efficacy in a breast cancer model near future.

## Figures and Tables

**Figure 1 pharmaceutics-11-00141-f001:**
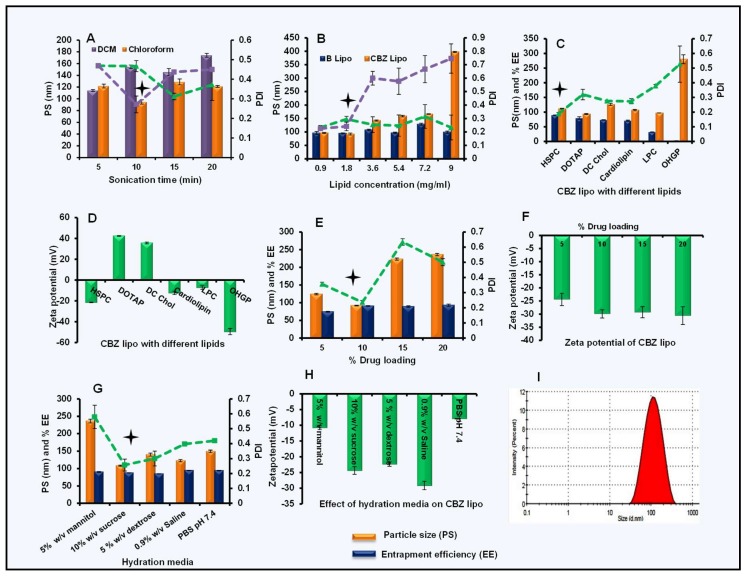
Preparation and optimization of cabazitaxel (CBZ) liposomes: **(A**) Effect of solvents on particle size and polydispersity index (PDI) of blank liposomes, (**B**) effect of lipid concentration on blank and CBZ liposomes, (**C**) effect of type of lipid on CBZ liposomes, (**D**) zeta potential of CBZ liposomes with different lipids, (**E**) effect of drug loading on PS, PDI, and percentage of entrapment efficiency (% EE), (**F**) zeta potential at varied drug loading, (**G**) effect of hydration media on CBZ liposomes, (**H**) zeta potential of CBZ liposomes with different hydration media, and (**I**) particle size distribution for optimized CBZ liposomes. Data was represented in mean ± SD.

**Figure 2 pharmaceutics-11-00141-f002:**
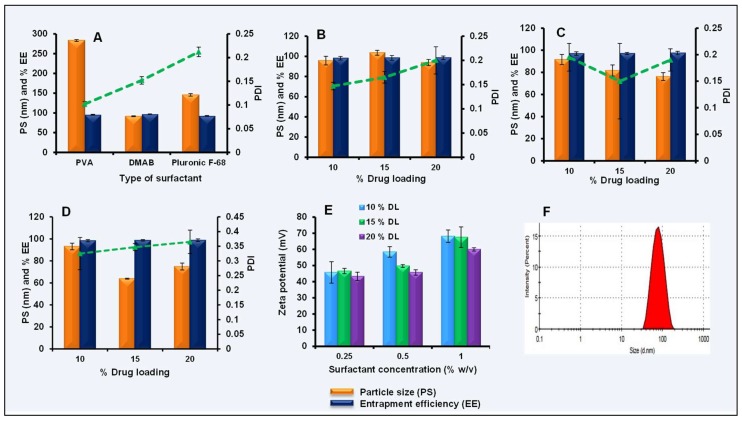
Preparation and optimization of CBZ nanoparticles (NPs): (**A**) Effect of type of surfactant on particle size, PDI, and entrapment efficiency of CBZ NPs, (**B**) effect of 0.25% *w*/*v* didodecyltrimethylammonium bromide (DMAB) concentration on particle size, PDI, and entrapment efficiency of CBZ NPs, (**C**) effect of 0.5% *w*/*v* DMAB concentration on particle size, PDI, and entrapment efficiency of CBZ NPs, (**D**) effect of 1% *w*/*v* DMAB concentration on particle size, PDI, and entrapment efficiency of CBZ NPs, (**E**) effect of surfactant concentration and drug loading on zeta potential of CBZ NPs, (**F**) particle size distribution for optimized CBZ NPs. Data was represented with mean ± SD.

**Figure 3 pharmaceutics-11-00141-f003:**
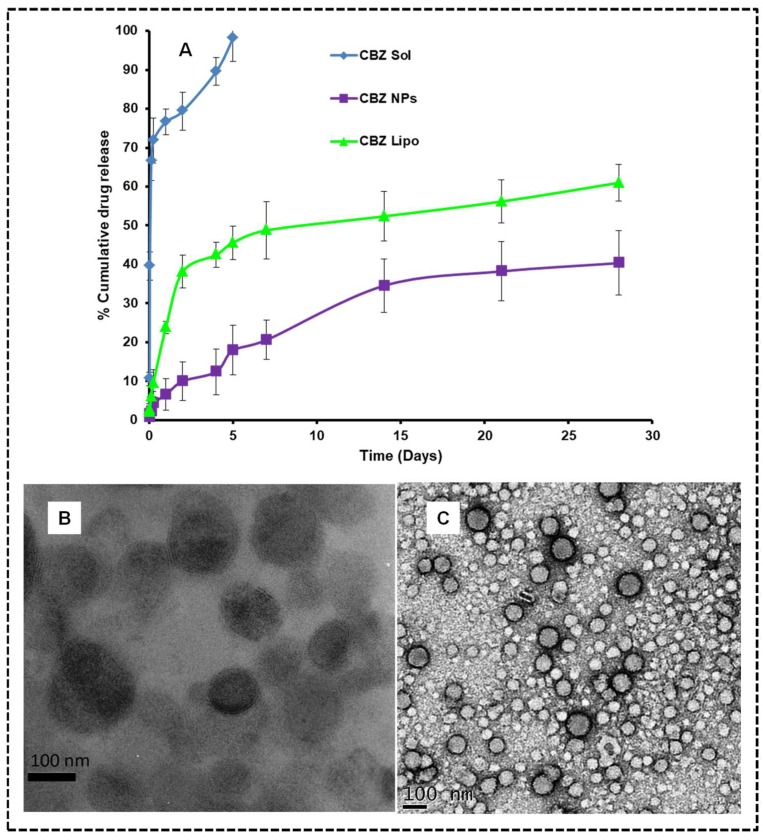
In vitro drug release and TEM images for optimized formulations: (**A**) The dialysis bag method was used to perform release studies in phosphate buffer saline (PBS) pH 7.4 (0.5% *w*/*v* tween 80) as a release media, and the data is presented as percentage of cumulative drug release vs. time graph. The data was represented in mean ± SD. (**B**) TEM image of CBZ liposomes and (**C**) TEM image of CBZ NPs in correlation with dynamic light scattering (DLS) results.

**Figure 4 pharmaceutics-11-00141-f004:**
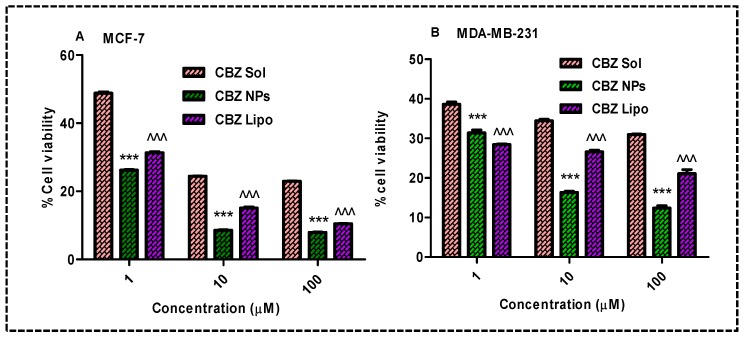
Cell viability assay: MCF-7 and MDA-MB-231 cells were treated with formulations for 48 h and the percentage of cell viability was calculated in comparison with control, (**A**) there was a significant decrease in the percentage of cell viability by an increase in the concentration of CBZ on MCF-7 cells after treatment with CBZ NPs and CBZ liposomes (*** *p* < 0.001 and ^^^ *p* < 0.001) in comparison with the CBZ micellar solution, respectively, (**B**) significant reduction in the percentage of cell viability by an increase in the concentration of CBZ on MDA-MB-231 cells after treatment with CBZ NPs and CBZ liposomes (*** *p* < 0.001 and ^^^ *p* < 0.001) in comparison with the CBZ solution respectively; the data was represented in mean ± SD.

**Figure 5 pharmaceutics-11-00141-f005:**
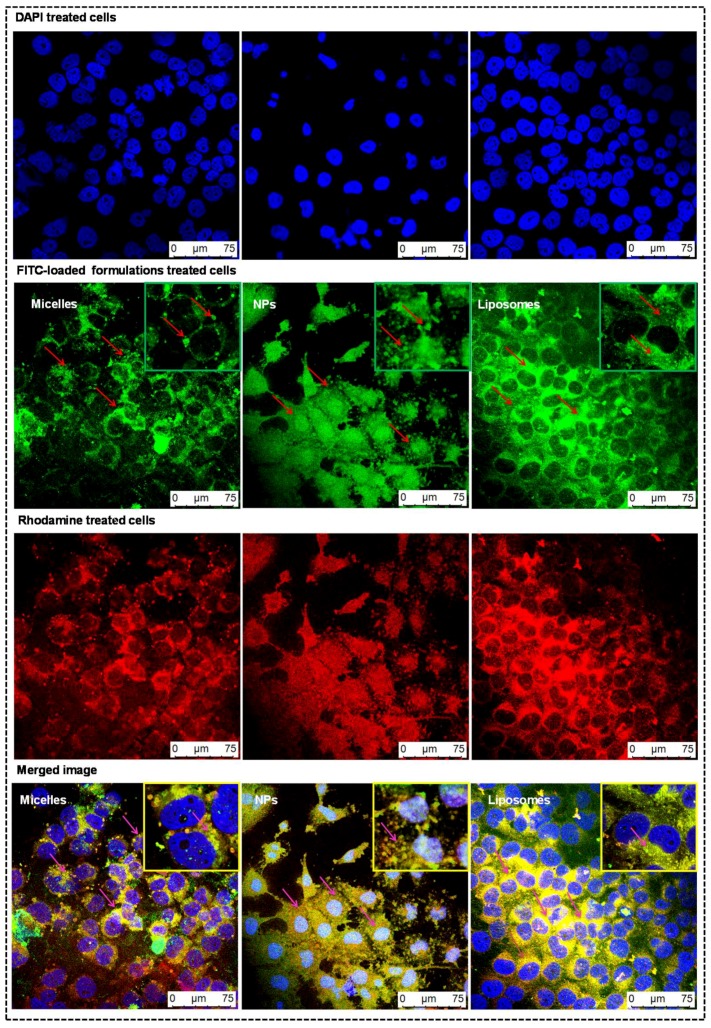
Qualitative cellular internalization of nanocarriers by confocal microscopy: MDA-MB-231 cells were treated with fluorescein isothiocyanate (FITC)-loaded formulations for 3 h to get green fluorescence in the cytoplasm of cells and with rhodamine to obtain red fluorescence. DAPI as a nuclear stain in cells will give blue fluorescence. The first row represents the cells treated with DAPI, which are colored blue. The second row represents the cells treated with FITC-loaded formulations, which are colored green. The third row represents the cells treated with rhodamine, which are colored red. The fourth row represents the merge image of above three rows, which were captured with confocal laser scanning microscopy (CLSM) at 400× magnification, and shows the colocalization of formulations in the cell membrane.

**Figure 6 pharmaceutics-11-00141-f006:**
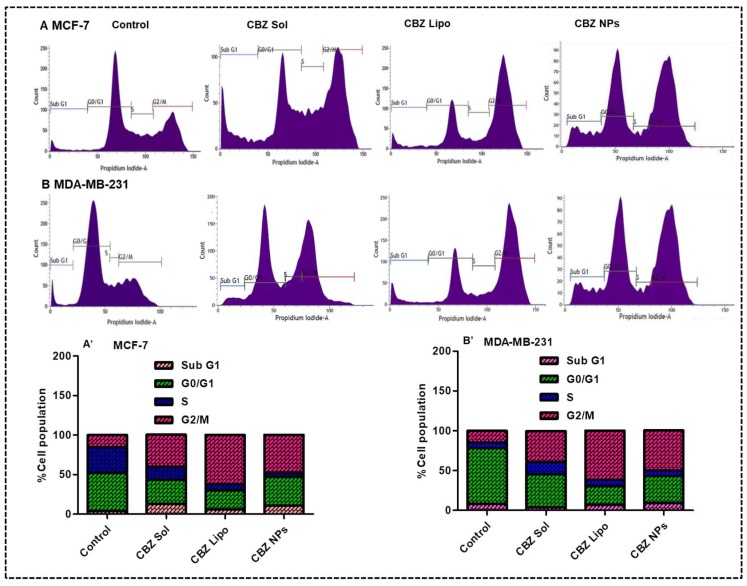
Inhibition of cell cycle and DNA damage checkpoints on MCF-7 and MDA-MB-231 cell lines: (**A**,**B**) representing flow cytograms of different phases of arrest in the cell cycle in MCF-7 and MDA-MB-231 cell lines treated with CBZ micellar solution, CBZ liposomes, and CBZ NPs, respectively, in comparison with control. (**A′**,**B′**) represent the percentage of cell population at different phases of arrest in MCF-7 and MDA-MB-231 cell lines treated with CBZ micellar solution, CBZ liposomes, and CBZ NPs, respectively, in comparison with control.

**Figure 7 pharmaceutics-11-00141-f007:**
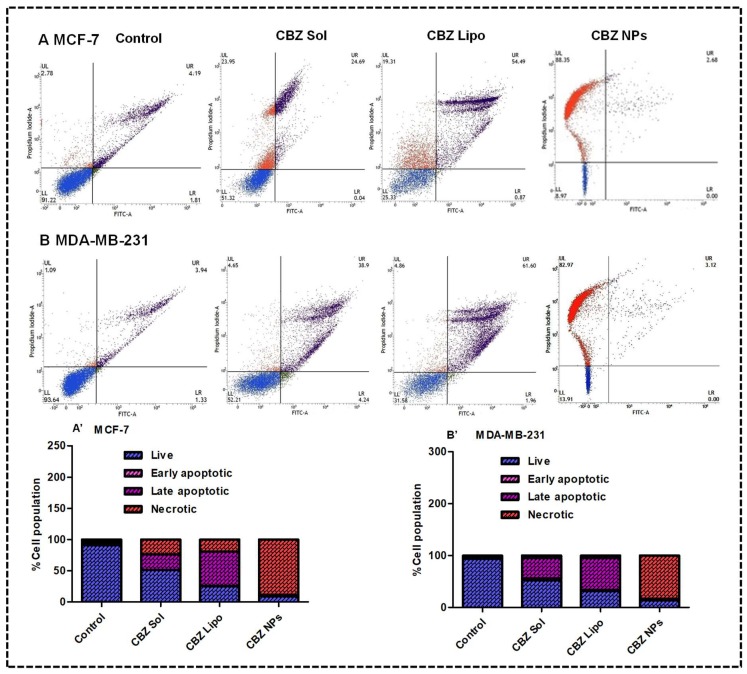
Cytometric detection of apoptosis on the MCF-7 and MDA-MB-231 cell lines: the Lower Left (LL) quadrant represents live cells, the Lower Right (LR) quadrant represents cells that are in the early apoptotic phase, the Upper Right (UR) quadrant represents cells that are in the late apoptotic phase, and the Upper Left (UL) quadrant represents cells that are in the necrotic phase (Dead). (**A**,**B**) demonstrate the flow cytograms of the MCF-7 and MDA-MB-231 cell lines after treatment with the CBZ micellar solution, CBZ liposomes, and CBZ NPs along with their cells at different quadrants in comparison with control, respectively. (**A′**,**B′**) demonstrated the percentage of cell population in the live, early apoptotic, late apoptotic, and necrotic phases.

**Figure 8 pharmaceutics-11-00141-f008:**
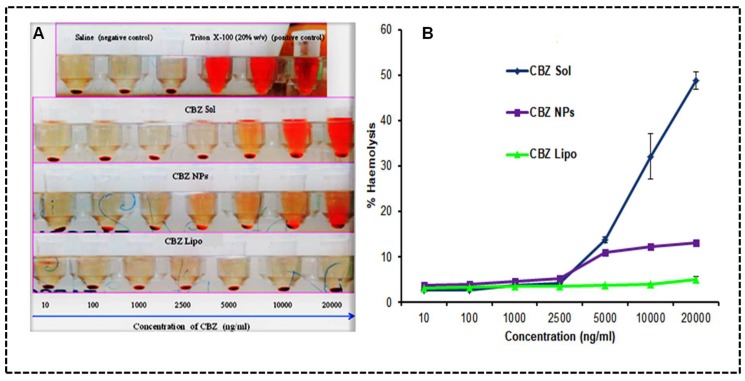
Ex vivo hemolysis: (**A**) Representation of red blood cells (RBCs) treated with CBZ micellar solution, CBZ NPs, and CBZ liposomes. (**B**) Percentage of hemolysis representing RBCs treated with the micellar CBZ solution, CBZ NPs, and CBZ liposomes. The data was represented in mean ± SD.

**Figure 9 pharmaceutics-11-00141-f009:**
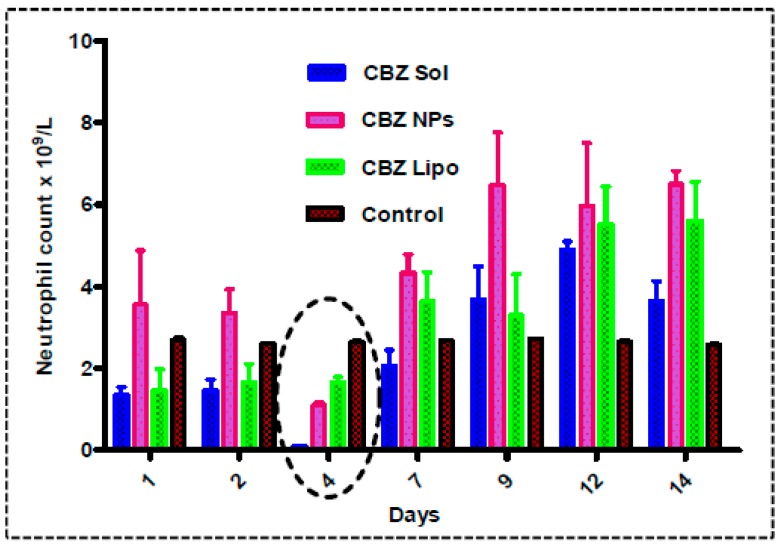
Neutropenic study: The graph representing neutrophil count until 14 days after the administration of CBZ micellar solution, CBZ NPs, and CBZ liposomes at 2.5 mg/kg dose in female Sprague–Dawley (SD) rats in comparison with saline control. The neutropenic effect in animals showed that there was a drastic reduction in neutrophil count (0.1025 × 10^9^/L) in animals administered with CBZ micellar solution on the fourth day. A depletion in neutrophil count was less observed in animals administered with CBZ liposomes (1.6725 × 10^9^/L) and CBZ NPs (1.1 × 10^9^/L) in comparison to CBZ micellar solution and saline control (2.635 × 10^9^/L).

**Figure 10 pharmaceutics-11-00141-f010:**
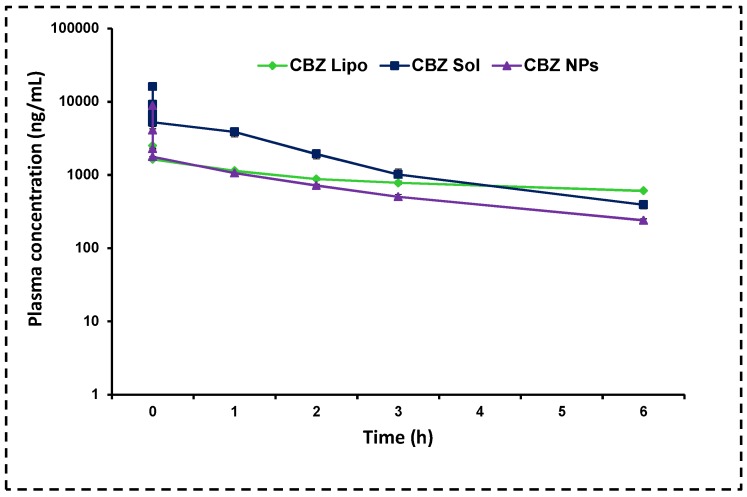
Pharmacokinetics: Mean plasma concentration and time profiles of micellar CBZ solution, CBZ liposomes, and CBZ NPs in rats after i.v. administration (dose 15 mg/kg). The results clearly showed that the micellar CBZ solution has shown more AUC_tot_ and *C*_max_ because of the presence of tween 80 and ethanol, which is lethal to cause neutropenia. In contrast, the groups administered CBZ NPs and CBZ liposomes have shown a 0.413-fold, 3.15-fold increase in half-life and 0.339-fold and 3.785-fold increase in MRT respectively. The data was represented in mean ± SD.

**Table 1 pharmaceutics-11-00141-t001:** Hematological screening for different formulations of CBZ on the fourth day (data as mean ± SE, *n* = 5).

Blood Parameters	Unit	Control	Micellar Solution	NPs	Liposomes
WBCP	×10^3^ cells/µL	14.67 ± 1.02	14.22 ± 1.43	14.57 ± 1.14	14.12 ± 3.11
WBCB	×10^3^ cells/µL	15.09 ± 1.18	14.65 ± 3.21	15.32 ± 1.51	14.94 ± 2.17
RBC	×10^6^ cells/µL	8.85 ± 0.95	6.82 ± 0.36	7.46 ± 2.04	8.12 ± 1.02
Neut	×10^9^ cells/L	2.63 ± 0.232	0.1025 ± 0.65	1.1053 ± 0.48	1.6725 ±1.02
HGB	g/dL	111.24 ± 3.27	10.30 ± 3.71	10.57 ± 1.27	11.07 ± 3.20
HCT	%	443.9 ± 1.12	40.9 ± 2.62	41.1 ± 1.22	43.1 ± 1.26
MCV	fL	557.81 ± 3.16	58.21 ± 1.17	56.43 ± 1.04	57.68 ± 1.02
MCH	Pg	114.12 ± 1.24	13.93 ± 2.12	14.23 ± 1.17	14.35 ± 1.32
MCHC	g/dL	224.35 ± 0.95	23.24 ± 0.30	24.15 ± 2.05	24.29 ± 1.22
CHCM	g/dL	235.1 ± 1.26	33.8 ± 2.45	33.7 ± 1.25	33.3 ± 1.55
RDW	%	115.60 ± 3.13	14.75 ± 3.15	15.45 ± 2.07	16.45 ± 1.16
HDW	g/dL	2.33 ± 0.54	2.14 ± 0.41	2.17 ± 0.32	2.14 ± 0.18
PLT	×10^3^ cells/µL	192 ± 12.35	155 ± 22.02	178 ± 24.34	232 ± 14.56
MPV	fL	314.12 ± 1.25	9.51 ± 3.12	9.83 ± 3.45	11.52 ± 5.23

WBCP—White blood cell count perox, WBCB—White blood cell count baso, HGB—Hemoglobin, HCT—Hematocrit, MCV—Mean corpuscular volume, MCH—Mean corpuscular hemoglobin, MCHC—Mean corpuscular hemoglobin concentration, CHCM—Cell hemoglobin concentration mean, RDW—Red cell distribution width, HDW—Hemoglobin distribution width, PLT—Platelets, MPV—Mean platelet volume.

**Table 2 pharmaceutics-11-00141-t002:** Pharmacokinetic parameters for different CBZ formulations with a dose of 15 mg/kg cabazitaxel after tail vein injection. Data are presented mean ± SD (*n* = 6).

Parameter	Micellar Solution	NPs	Liposomes
*C*_max_ (ng/mL)	16,075.5 ± 4.12	8858.05 ± 2.02 ***	7959.3 ± 1.07 ***
*T*_max_ (h)	0.083 ± 0.05	0.083 ± 0.07	0.083 ± 0.05
AUC_tot_ (ng·h/L)	15,008.42 ± 12.51	6902.48 ± 2.15 ***	14,223.0 ± 2.07 ***
*t*_1/2_ (h)	1.828 ± 0.05	2.583 ± 0.12 ***	7.585 ± 0.09 ***
MRT (h)	1.856 ± 0.07	2.485 ± 0.05 ***	8.882 ± 0.17 ***

*C*_max_: maximum peak plasma concentration, *T*_max_: time at which the peak plasma concentration was obtained, AUC_tot_: total area under the curve after drug absorption, *t*_1/2_: half-life of the drug, MRT: mean residence time of the drug. Micellar solution vs. NPs, micellar solution vs. liposomes, and NPs vs. liposomes (*** *p* < 0.001).

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
