# Peer review of "Cabazitaxel-Loaded Nanocarriers for Cancer Therapy with Reduced Side Effects"

_pharmaceutics, 2019, doi:10.3390/pharmaceutics11030141_

Round 1

Reviewer 1 Report

This is an extensive work by Dr. Khan group. This study reports minimze toxicity associated with micellar formulation of cabazitaxel which is commonly used anticancer drug for various therapeutic applications including breast cancer. This study evaluated in vitro primarily but now in vivo. It is highly encouraged to show improved therapeutic benefit with this formulation would be a wonderful article. However, in its current form, i have few suggestions to authors:

Cell viability assay: Control micelle, lipo, and nanoparticles are needed to compare with treatment groups. what i see the additional toxicity might be arising from nanoparticles?

Figure 5, cellular internalization: It is believed that dye will be released from micelle, nanoparticles and liposomes in that case how it can be compared with one another? Quantification is needed for this figure. Also flow cytometry data is more appropriate for these studies.

Pharmacokinetics: It is a good data according to the manuscript but how this is going to translatable for therapeutic studies? Authors do not have data to show these formulations may show better or at least equivalent therapeutic benefit?

Author Response

Reviewer 1:

This is an extensive work by Dr. Khan group. This study reports minimize toxicity associated with micellar formulation of cabazitaxel which is commonly used anticancer drug for various therapeutic applications including breast cancer. This study evaluated in vitro primarily but now in vivo. It is highly encouraged to show improved therapeutic benefit with this formulation would be a wonderful article. However, in its current form, i have few suggestions to authors:

We appreciate reviewer for recognizing the worth of manuscript

Comment 1: Cell viability assay: Control micelle, lipo, and nanoparticles are needed to compare with treatment groups. What I see the additional toxicity might be arising from nanoparticles?

Response: Primarily blank formulations were screened and the % cell viability was found to be nearly 100 % which was mentioned in the manuscript section 3.5, page no.11, and line number 402. Blank nanoparticles have not shown any cytotoxicity as they were prepared with PLGA which is a biodegradable FDA approved polymer and different surfactants. Before treatment to the cell lines the nanoparticles were prewashedto remove surfactant. These results are also in correlation with our previous published article form our lab (1). Thus we can say that the additional toxicity arising from nanoparticles is due to Cabazitaxel loaded in nanoparticles

(1) Muntimadugu E, Kumar R, Saladi S, Rafeeqi TA, Khan W. CD44 targeted chemotherapy for co-eradication of breast cancer stem cells and cancer cells using polymeric nanoparticles of salinomycin and paclitaxel. Colloids Surf B Biointerfaces. 2016;143:532-46.

Comment 2: Figure 5, cellular internalization: It is believed that dye will be released from micelle, nanoparticles and liposomes in that case how it can be compared with one another? Quantification is needed for this figure. Also flow cytometry data is more appropriate for these studies.

Response: We appreciate the reviewer for this question. The dye (FITC) loaded in the nanocarriers was hydrophobic as cabazitaxel and won’t show immediate release and need days to release as observed for cabazitaxel in figure 3A page number 10. These uptake experiments were conducted only for 3 h. After treating with dye loaded nanocarriers, first in initial time of 3 h cellular internalization will takes place by endocytosis, there will be very less or no release of dye from the nanocarriers which was observed in confocal microscopy by qualitative estimation  as shown in figure 5. Flow cytometric estimation will be more appropriate for this experiment, but these are only to show uptake of nanocarriers and internalized to kill the cancer cells.

Comment 3: Pharmacokinetics: It is a good data according to the manuscript but how this is going to translatable for therapeutic studies? Authors do not have data to show these formulations may show better or at least equivalent therapeutic benefit?

Response: We appreciate the reviewer for recognizing the worth of the data in the manuscript. This paper is for formulation development and in vitro studies. It’s in vivo and clinical translation will be surely followed in further work.

Reviewer 2 Report

Comments and Suggestions for Author

Manuscript: Cabazitaxel loaded nanocarriers for cancer therapy with reduced side effects” by Nagavendra Kommineni, Shaheen Mahira, Abraham J. Domb and Wahid Khan, is an interesting and well-written manuscript, that covers well-designed studies on the development of surfactant free nanoformulations of cabazitaxel identifying the therapeutic potential of cabazitaxel loaded nanocarriers for treatment of breast cancer cell lines. However, some corrections are essential before publication.

Line 159:  How do you know the total drug incorporated in 1 mL of liposome or NPs dispersion? 

Line 182: Could you add the reference for Higuchi and Peppas models to study the drug release kinetics? 

Line 199: In the assays of cellular internalization of nanocarriers by confocal microscopy, was only included experiments with the cell line MDA-MB-231, could you explain why not included the experiments with MCF-7?

Line 304: You mentioned, “Particle should have neutral or slightly negative zeta potential to minimize nonspecific binding to endothelium”. What is the range of this value?

Line 367: In the TEM analysis, you should include the size (average) and compare it with the hydrodynamic size measured by dynamic light scattering. 

In the figure 10, the plasma concentrations of CBZ NPs and CBZ Lip appear to be similar; however, in the table 2, it is observed that there was a significant difference in t½ and MRT between both formulations (t½ 2.583 h to CBZ NPs Vs 7.585 h to CBZ Lip; and MRT 2.485 h to CBZ NPs Vs 8.882 h to CBZ Lip); the authors could explain this discrepancy of the data?, and it would be better to show the graphic in logarithmic scale?, and to corroborate the paragraph of the Line 557:  “Drastic improvement in t½ and mean residence time of CBZ liposomes over CBZ NPs is due to presence of DSPE-PEG 2000 which prolongs the circulation and avoid macrophage uptake which is in correlation with earlier reports”.  

It is necessary to show the statistical analysis for Hematological screening (table 1), and pharmacokinetic parameters (Table 2).

It would have been interesting and important to include data of in-vivo models of breast cancer in order to evaluate the systemic efficacy of the proposed NPs and liposomal systems. Although the inclusion of those studies is not mandatory for this publication, it is suggested to include this point as a perspective of the work, or include a short statement “What other studies are planned for the future or which are the next steps” for these nanoformulations?

Author Response

Reviewer 2:

Manuscript: Cabazitaxel loaded nanocarriers for cancer therapy with reduced side effects” by Nagavendra Kommineni, Shaheen Mahira, Abraham J. Domb and Wahid Khan, is an interesting and well-written manuscript, that covers well-designed studies on the development of surfactant free nanoformulations of cabazitaxel identifying the therapeutic potential of cabazitaxel loaded nanocarriers for treatment of breast cancer cell lines. However, some corrections are essential before publication.

We appreciate reviewer for recognizing the worth of manuscript

Comment 1: Line 159:  How do you know the total drug incorporated in 1 mL of liposome or NPs dispersion? 

Response: First we took known amount of drug for preparation of liposomes and nanoparticles then we have determined total formulation volume obtained in experiment. Then we took 1 ml and determined drug content by procedure given in section 2.5.2 and page number 4. Finally we multiplied the total volume with drug present in 1 ml to get total drug encapsulated in liposomes and nanoparticles.

Comment 2: Line 182: Could you add the reference for Higuchi and Peppas models to study the drug release kinetics? 

Response: Kindly see reference cited in 28, page number 20, highlighted in yellow lines.

Comment 3: Line 199: In the assays of cellular internalization of nanocarriers by confocal microscopy, was only included experiments with the cell line MDA-MB-231, could you explain why not included the experiments with MCF-7?

Response: MDA-MB-231 cell lines are resistant and triple negative breast cancer cell lines whereas MCF-7 cell lines are sensitive and hormone receptor positive as mentioned in section 2.7, page number 5. Cellular internalization of nanocarriers is mostly by endocytosis through cell membranes of cancer cells. As both the cell lines represent breast cancer, hence, the study in most resistant cell line like MDA-MB-231 is presented as representation.

Comment 4: Line 304: You mentioned, “Particle should have neutral or slightly negative zeta potential to minimize nonspecific binding to endothelium”. What is the range of this value?

Response: As per the previous report (1) we mentioned this statement.  Another article mentioned ± 40 mV (2). This has been mentioned in line 310.

1. Dawidczyk CM, Kim C, Park JH, Russell LM, Lee KH, Pomper MG, et al. State-of-the-art in design rules for drug delivery platforms: lessons learned from FDA-approved nanomedicines. Journal of Controlled Release. 2014;187:133-44.

2. Honary S, Zahir F. Effect of zeta potential on the properties of nano-drug delivery systems-a review (Part 1). Tropical Journal of Pharmaceutical Research. 2013;12(2):255-64.

Comment 5:  Line 367: In the TEM analysis, you should include the size (average) and compare it with the hydrodynamic size measured by dynamic light scattering. 

Response: TEM is performed to determining surface morphology and particle shape. Still there is scale bar (100 nm) for both liposomes and nanoparticles to give representative particle size range. For determining size dynamic light scattering method was used which is most acceptable method to determine particle size. Please kindly refer figure 3, page number 10.

Comment 6:  In the figure 10, the plasma concentrations of CBZ NPs and CBZ Lip appear to be similar; however, in the table 2, it is observed that there was a significant difference in t½ and MRT between both formulations (t½ 2.583 h to CBZ NPs Vs 7.585 h to CBZ Lip; and MRT 2.485 h to CBZ NPs Vs 8.882 h to CBZ Lip); the authors could explain this discrepancy of the data?, and it would be better to show the graphic in logarithmic scale?, and to corroborate the paragraph of the Line 557:  “Drastic improvement in t½ and mean residence time of CBZ liposomes over CBZ NPs is due to presence of DSPE-PEG 2000 which prolongs the circulation and avoid macrophage uptake which is in correlation with earlier reports”.

Response: To clarify this in figure 10, the plasma concentrations of rats injected with CBZ sol, CBZ NPs we could not able to detect the drug after 6 h time point. However in case of rats injected with CBZ Lipo we could able to detect the drug in plasma at 24 and 48 h time point. In order to compare between the three groups we kept the plasma concentration and time profile plot till 6 h. As per the suggestion we changed the graph with logarithmic scale and we could able to see the difference between in plasma concentration time profile graph of CBZ Lipo and CBZ NPs as shown in page number 18. We calculated the data again using Kinetica software; we are surprised to get the same results. There was a difference in plasma concentration from 2 h to 6 h time point between CBZ Lipo and CBZ NPs which has been impacted to get significant difference in t½ and MRT of CBZ NPs and CBZ Lipo. Drastic improvement in t½ and mean residence time of CBZ liposomes over CBZ NPs is due to presence of DSPE-PEG 2000 which prolongs the circulation and avoid macrophage uptake.  

Comment 7: It is necessary to show the statistical analysis for Hematological screening (table 1), and pharmacokinetic parameters (Table 2).

Response: We applied statistics for hematological screening for table 1. But we could not able to get significant difference in data as mentioned in manuscript line number 532-533. However, there was a difference only in neutrophil count in treatment groups as the cabazitaxel micellar solution causes neutropenia as side effect which was in correlation with previous reports by starpharma. We have included statistical analysis for pharmacokinetic parameters as mentioned in table 2 line number 574.

Starpharma. Starpharma’s DEP™ eliminates cabazitaxel neutropenia 2016 [cited 2017 25th July]. Available from: http://www.starpharma.com/news/284

Comment 8: It would have been interesting and important to include data of in-vivo models of breast cancer in order to evaluate the systemic efficacy of the proposed NPs and liposomal systems. Although the inclusion of those studies is not mandatory for this publication, it is suggested to include this point as a perspective of the work, or include a short statement “What other studies are planned for the future or which are the next steps” for these nanoformulations?

Response: We appreciate the reviewer for his valuable suggestion for in-vivo tumor model for breast cancer, which was more appropriate to determine the efficacy of the developed NPs and liposomes. The developed nanocarriers will be evaluated for their efficacy in breast cancer model is under progress. A statement was mentioned in page number 19 and line 610-611. 

Reviewer 3 Report

The manuscript “Cabazitaxel loaded nanocarriers for cancer therapy with reduced side effects” analyzes, through a set of different experimental techniques including DLS and TEM, the physical properties and drug entrapment efficiency of CBZ liposomes and nanoparticles, in comparison to the FDA approved micellar solution. Further in vitro and in vivo experiments evaluate these nanocarriers as antitumoral agents. The study highlights the importance of drug formulation into the adequate nanocarrier to accomplish the highest antitumor response minimizing the side effects of the drug. 

I suggest the publishing of this manuscript as Original Article in Pharmaceutics under Minor Revisions, including a thorough language check, which are listed below:

Results and discussion.

1.       During lipid concentration optimization, a big increase in particle size is observed for CBZ liposomes after using 9 mg/ml of HSPC. How do the authors explain this behavior?

2.       An interesting sustained release of CBZ is observed for nanoparticles and liposomes, accomplishing up to 40 and 60% release after 28 days. Nevertheless, pharmacokinetic study indicates that the mean residence time of the drug is 2.4 and 8.9 h, respectively, highlighting the need to evaluate the carriers´ performance at much shorter times. Could the authors expand this discussion?

3.       Increasing the nanocarrier concentration from 1 to 10 µM produces a substantial decrease on tumor cell viability, while the difference between 10 and 100 µM is barely noticeable. According to many previous studies on nanocarriers, 100 µM is a pretty high concentration which might be toxic to healthy tissues. How do the authors justify their choice for further experiments? Furthermore, similar viability studies using a control (non-tumor) cell line would be desirable.  

4.       The authors mentioned the use of rhodamine as cell membrane stain as well as cytoplasm stain. Please clarify the use of rhodamine in the experiment.

5.       Figures 6, 7 and 8 present duplicated information. The relevant data are represented in the diagrams. I suggest transferring the flow cytograms as well as the hemolysis photographs to the Supporting Information.

6.       According to the authors, the hematological screening presented in Table 1 does not present significant differences between the different nanocarriers. I suggest including this Table in the Supporting Information.

7.       The authors should include further discussion on the higher hemolytic behavior of CBZ nanoparticles, in comparison to CBZ liposomes.

Misspelling and language mistakes (some examples).

8.       Abstract: (line 9) “…cabazitaxel (CBZ) solution WAS approved…”; (line 10) “The purpose of this study is TO EVALUATE liposomes…”;

9.       Introduction: (line 42) “… there are remedies for the DISEASE,”; (line 47) “There is a need of MORE appropriate strategies…”; (line 65) “… to parent compound DTX due to…”;

10.   Results and discussion: (lines 381-382) “… FOLLOWED… 

Author Response

Reviewer 3:

The manuscript “Cabazitaxel loaded nanocarriers for cancer therapy with reduced side effects” analyzes, through a set of different experimental techniques including DLS and TEM, the physical properties and drug entrapment efficiency of CBZ liposomes and nanoparticles, in comparison to the FDA approved micellar solution. Further in vitro and in vivo experiments evaluate these nanocarriers as antitumoral agents. The study highlights the importance of drug formulation into the adequate nanocarrier to accomplish the highest antitumor response minimizing the side effects of the drug. 

I suggest the publishing of this manuscript as Original Article in Pharmaceutics under Minor Revisions, including a thorough language check, which are listed below:

We appreciate reviewer for recognizing the worth of manuscript

Comment 1: During lipid concentration optimization, a big increase in particle size is observed for CBZ liposomes after using 9 mg/ml of HSPC. How do the authors explain this behavior?

Response: As increase in lipid (HSPC) concentration to 9 mg/ml, proportionately we increased the CBZ concentration in the formulation. However, volume of the hydration media was same. As CBZ was highly hydrophobic there was a rapid precipitation lead to big increase in particle size was observed.

Comment 2: An interesting sustained release of CBZ is observed for nanoparticles and liposomes, accomplishing up to 40 and 60% release after 28 days. Nevertheless, pharmacokinetic study indicates that the mean residence time of the drug is 2.4 and 8.9 h, respectively, highlighting the need to evaluate the carriers´ performance at much shorter times. Could the authors expand this discussion?

Response: As the drug (CBZ) is highly hydrophobic under BCS class IV with limited solubility in aqueous media. In order to enhance the solubility of drug released from nanocarriers we used 0.5 % w/v tween 80 in phosphate buffer saline. Hence there was a sustained drug release from polymer matrix and liposomes. However in pharmacokinetic studies after intravenous administration, there are several parameters which will influence the drug distribution and elimination like presence of plasma proteins, CYP enzymes, elimination and metabolism kinetics by kidney and liver. MRT of the formulation depends on above all the parameters. So, in-vitro and in-vivo correlation is not much easy for nanocarriers. This was included in the conclusions section 4, page number 19.

Comment 3: Increasing the nanocarrier concentration from 1 to 10 µM produces a substantial decrease on tumor cell viability, while the difference between 10 and 100 µM is barely noticeable. According to many previous studies on nanocarriers, 100 µM is a pretty high concentration which might be toxic to healthy tissues. How do the authors justify their choice for further experiments? Furthermore, similar viability studies using a control (non-tumor) cell line would be desirable.  

Response: We appreciate the reviewer for this question. Increasing the nanocarrier concentration from 1 to 10 µM produces a substantial decrease on tumor cell viability, while the difference between 10 and 100 µM is barely noticeable. According to previous reports, taxane based chemotherapeutics as potent microtubule inhibitor doesn’t shown concentration dependent cytotoxicity at higher concentrations. However, they show cytostatic effect in cancer cell lines. We observed that at 100 µM concentration we got better cancer cell killing, the concentration which we have selected in correlation with previous article published as mentioned below. We carried out the other experiments based on this evidence and we got promising results. There is no meaning of doing cell viability studies on non-tumor cell lines because the potent anticancer molecules loaded in nanocarriers will show cell death at low concentrations.

Mahdaviani P, Bahadorikhalili S, Navaei-Nigjeh M, Vafaei SY, Esfandyari-Manesh M, Abdolghaffari AH, et al. Peptide functionalized poly ethylene glycol-poly caprolactone nanomicelles for specific cabazitaxel delivery to metastatic breast cancer cells. Mater Sci Eng C. 2017;80:301-12.

Comment 4: The authors mentioned the use of rhodamine as cell membrane stain as well as cytoplasm stain. Please clarify the use of rhodamine in the experiment.

Response: Rhodamine is hydrophilic dye which was used as tracer dye to determine the rate and direction of flow and transport of nanocarriers. Similar experiment was performed with in previous studies as reported below.

Zhang Y, Hensel M. Evaluation of nanoparticles as endocytic tracers in cellular microbiology. Nanoscale. 2013;5(19):9296-309.

Di Paolo M, Roberti MJ, Bordoni AV, Aramendía PF, Wolosiuk A, Bossi ML. Nanoporous silica nanoparticles functionalized with a fluorescent turn-on spirorhodamineamide as pH indicators. Photochemical & Photobiological Sciences. 2019;18(1):155-65.

Bulbake U, Kommineni N, Ionov M, Bryszewska M, Khan W. Comparison of Cationic Liposome and PAMAM Dendrimer for Delivery of Anti-Plk1 siRNA in Breast Cancer Treatment. Pharmaceutical development and technology. 2019 (just-accepted):1-27.

Comment 5: Figures 6, 7 and 8 present duplicated information. The relevant data are represented in the diagrams. I suggest transferring the flow cytograms as well as the hemolysis photographs to the Supporting Information.

Response: We apologize the reviewer for this question. The figures presented are not duplicated. There is a need of keen observation in flow cryptograms with difference in cell scattering at different phases. As we used single (100 μM) concentration of CBZ sol, CBZ NPs and CBZ Liposomes, we got difference in cell death. As per the suggestion we are providing the supporting data for figures as mentioned below:

Supporting data for figure 6

MCF-7

Sub G1

     G0/G1

S

G2/M

Control

3.97±0.31

48.30±0.52

32.46±0.22

15.45±1.12

CBZ Sol

13.09±1.02

30.54±2.14

16.62±3.01

40.25±1.17

CBZ Lipo

6.00±0.15

23.99±0.53

8.41±1.07

61.84±2.09

CBZ NPs

11.16±0.23

35.88±0.42

5.31±0.55

48.03±1.32

Inhibition of   cell cycle and DNA damage checkpoints

MDA-MB-231

Sub G1

G0/G1

S

G2/M

Control

8.32±0.08

70.05±1.03

7.01±0.32

14.95±0.47

CBZ Sol

3.99±0.16

41.65±1.32

15.18±0.40

38.94±0.11

CBZ Lipo

7.02±0.13

23.32±0.62

8.43±0.32

61.49±0.77

CBZ NPs

9.36±0.81

34.11±1.02

6.99±0.09

50.12±2.01

Supporting data for figure 7

MCF-7

Live  phase

Early   apoptotic phase

Late   apoptotic phase

Necrotic   phase

Control

91.22±0.12

1.81±0.22

4.19±0.15

2.78±0.19

CBZ Sol

51.32±0.23

0.04±0.18

24.69±1.12

23.95±2.02

CBZ Lipo

25.33±0.10

0.87±0.32

54.49±1.25

19.31±0.18

CBZ NPs

8.97±0.31

0.00±0.05

2.68±0.08

88.35±3.62

Cytometric   detection of apoptosis

MDA-MB-231

Live   phase

Early   apoptotic phase

Late   apoptotic phase

Necrotic   phase

Control

93.64±0.16

1.33±0.43

3.94±1.12

1.09±0.09

CBZ Sol

52.21±0.13

4.24±0.21

38.90±0.27

4.65±0.04

CBZ Lipo

31.58±0.12

1.96±0.33

61.60±0.41

4.86±0.11

CBZ NPs

13.91±0.24

0.00±0.03

3.12±0.10

82.97±2.31

Supporting data for figure 8

%   Hemolysis for CBZ  nano-formulations

Concentration of CBZ (ng/mL)

CBZ Sol ± SD

CBZ   Lipo ± SD

CBZ   NPs ± SD

10

2.77±0.07

3.75±0.10

3.14±0.16

100

2.80±0.12

4.07±0.12

3.32±0.12

1000

3.84±0.30

4.68±0.16

3.50±0.62

2500

4.24±0.23

5.21±0.10

3.68±0.68

5000

13.80±0.59

11.03±0.11

3.89±0.51

10000

32.11±5.00

12.24±0.21

4.10±0.28

20000

48.88±1.86

13.10±0.59

5.03±0.57

Comment 6: According to the authors, the hematological screening presented in Table 1 does not present significant differences between the different nanocarriers. I suggest including this Table in the Supporting Information.

Response: We agree with the results in hematological screening there was no significant difference between nanocarriers. However our intention is to compare with micellar solution, there we could able to get difference. We have seen there was a drastic depletion in neutrophil count on 4th day by cabazitaxel micellar solution which was less in case of liposomes and nanocarriers. The results are in correlation with previous reports mentioned by starpharma as given below.

Starpharma. Starpharma’s DEP™ eliminates cabazitaxel neutropenia 2016 [cited 2017 25th July]. Available from: http://www.starpharma.com/news/284.

Comment 7: The authors should include further discussion on the higher hemolytic behavior of CBZ nanoparticles, in comparison to CBZ liposomes.

ResponseAs per the suggestion further discussion was included in section 3.9 line numbers 515-519, page number 15. CBZ liposomes and CBZ NPs have shown 8.7 fold, 2.7 fold reduction in hemolytic potential in comparison to CBZ micellar solution and they are safe, biodegradable and biocompatible to administer systemically when compared to surfactant based CBZ micellar solution. Further CBZ liposomes have shown 1.6 fold reduction in hemolysis in comparison to CBZ NPs at 20 μg/mL.

Misspelling and language mistakes (some examples).

We appreciate the reviewer for his valuable suggestion which will enhance the quality of our manuscript.

Comment 8: Abstract: (line 9) “…cabazitaxel (CBZ) solution WAS approved…”; (line 10) “The purpose of this study is TO EVALUATE liposomes…”;

Response: Suggested changes were made in the abstract (line 9 and 10) and highlighted in yellow color.

Comment 9: Introduction: (line 42) “… there are remedies for the DISEASE,” (line 47) “There is a need of MORE appropriate strategies…”; (line 65) “… to parent compound DTX due to…”;

Response: Suggested changes were made in the abstract (line 42, 47 and 65) and highlighted in yellow color.

Comment 10: Results and discussion: (lines 381-382) “… FOLLOWED… 

Response: Suggested changes were made in line 383 highlighted in yellow colors.

Round 2

Reviewer 1 Report

I observe amendment of all queries related information in this revision.